# Analysis of Cytosine Base Editors in Bovine Zygotes: Efficiency and Editing Window Characterization Through Targeting the *MYO7A* Gene

**DOI:** 10.3390/cimb47121033

**Published:** 2025-12-11

**Authors:** Junghyun Ryu, Rebecca Tippner-Hedges, Martha Neuringer, Jon D. Hennebold

**Affiliations:** 1Division of Reproductive & Developmental Sciences, Oregon National Primate Research Center, Oregon Health & Science University, Beaverton, OR 97006, USA; ryuj@ohsu.edu (J.R.); tippnerh@ohsu.edu (R.T.-H.); 2Division of Neuroscience, Oregon National Primate Research Center, Oregon Health & Science University, Beaverton, OR 97006, USA; neuringe@ohsu.edu; 3Casey Eye Institute, Oregon Health & Science University, Portland, OR 97239, USA; 4Department of Obstetrics & Gynecology, Oregon Health & Science University, Portland, OR 97239, USA

**Keywords:** cytosine base editing, gene editing, Usher syndrome

## Abstract

Cytosine base editors (CBEs) enable precise C-to-T (G-to-A) conversions in genomic DNA, offering significant potential for specific gene editing. This study compared the prototypical Base Editor 3 (BE3) and a modified variant, BE3-Y130F, which utilizes an hA3A deaminase with the Y130F mutation, focusing on their editing efficiency and editing window characteristics using bovine zygotes. Following in vitro fertilization (IVF), sgRNA and Cas9 mRNA were injected as a targeting efficiency control, which resulted in 100% editing with no wild-type sequence. Then, either BE3 or BE3-Y130F mRNA, synthesized via in vitro transcription, and an sgRNA targeting exon 4 of the *MYO7A* gene was injected into zygotes. Genomic DNA was extracted from both blastocysts and developmentally arrested embryos, and Sanger sequencing was performed to evaluate C-to-T conversion efficiency and editing window. Both BE3 and BE3-Y130F achieved 100% C-to-T conversion efficiency at the primary target cytosine. BE3 displayed a defined editing window, primarily affecting cytosines at positions 7 and 8, indicating a predictable profile. In contrast, BE3-Y130F maintained high efficiency but had a less clearly defined editing window, resulting in incomplete editing and a remaining cytosine on the target sequence.

## 1. Introduction

The generation of genetically modified domesticated animal species and nonhuman primates is essential for modeling human diseases and developing novel therapeutics. However, producing these models is challenging. Somatic cell nuclear transfer (SCNT) has a low efficiency in primates, making it a problematic approach for disease model production [1,2]. As an alternative, direct cytoplasmic injection of CRISPR/Cas9 into zygotes has become a more reliable method. While this technique can generate gene knockouts, it frequently introduces random insertions or deletions (indels) at the target site [3]. The CRISPR/Cas9 system also has issues with mosaicism and off-target editing, which can lead to complex genotypes in animals and unintended gene modifications resulting in unexpected phenotypes [4].

A significant problem with this approach is the creation of triplet indel mutations, indels where the number of inserted or deleted bases is a multiple of three. These mutations do not cause a frameshift and therefore may not create a premature stop codon. Instead, they can result in a hypomorphic phenotype, where the target gene maintains partial function [5]. This outcome complicates the analysis of the disease model and, crucially, often renders the valuable, hard-to-obtain primate embryos unusable for the intended research. Given the significant cost, effort, and ethical considerations involved in producing rhesus macaque embryos, minimizing the generation of unusable blastocysts is a critical priority.

To address this limitation, base editing technology offers a more precise and efficient solution. Base editors can directly convert a single target DNA base to another without inducing double-strand breaks (DSBs), thereby avoiding the unpredictable indel patterns associated with CRISPR/Cas9 [6]. Specifically, cytosine base editors (CBEs) can be programmed to convert a cytosine to thymine (guanine to adenine) base pair. This is achieved by fusing a Cas9 nickase (Cas9n), which nicks the non-edited strand, to a cytidine deaminase enzyme. Guided by an sgRNA, the editor targets a specific DNA locus where the deaminase converts cytosine (C) to uracil (U). DNA repair machinery then resolves the resulting U:G mismatch into a T:A pair. This process can be precisely targeted to create a premature stop codon by converting CAA, CAG, or CGA codons to TAA, TAG, or TGA, ensuring a complete loss-of-function phenotype with high efficiency.

This study was designed to evaluate the potential of base editing as a superior tool for generating gene knockout models using bovine embryos. An advantage of bovine embryos is that they are readily available commercially and more closely resemble the fertilization and embryo development processes of humans compared to rodent models. Bovine systems are particularly valuable for pre-clinical refinement of gene-editing and microinjection techniques, as insights from in vitro embryo manipulation directly inform primate IVF protocols and regulatory pathways in early development [7]. We compared the performance of two different cytosine base editors: the original Base Editor 3 (BE3), which uses the human APOBEC3A (hA3A) deaminase, and a modified variant, BE3-Y130F, which utilizes a hA3A deaminase containing the Y130F modification, which was designed to increase substrate specificity and reduce off-target deamination compared with BE3 [8]. BE3 was selected due to its popularity in recent base-editing applications for its expanded editing window (2–13) and superior efficiency in GC-rich regions compared to APOBEC1-based predecessors [9]. By injecting the mRNA for each editor into bovine zygotes, we aimed to characterize and compare their C-to-T conversion efficiency and editing window. The *MYO7A* locus was targeted to recapitulate disruptions in Usher Syndrome type 1B, a sensory disorder we aim to model in rhesus macaques in the future by introducing premature stop codons in this conserved motor protein gene. Our results show that both editors achieved 100% editing efficiency, but BE3 demonstrated a more consistent and predictable editing window, providing valuable information for its future application in generating primate disease models in combination with assisted reproductive technologies.

## 2. Materials and Methods

### 2.1. Injection Material Preparation

To disrupt *MYO7A* exon 4 which contains a critical ATP binding domain, an sgRNA (5′-TCCACGCCGTGGACTGAAGT-3′) was designed using web-based tools CRISPOR [10]. The design considered the on-target efficiency score and position of cytosine within sgRNA (Figure 1). Selected sgRNAs were purchased from (SyntheogoMenlo Park, CA, USA). Plasmids for Cas9 (Addgene, Cambridge, MA, USA, #42230), BE3 (Addgene #113410), and BE3-Y130F (Addgene #113428) were obtained from Addgene [11]. To produce mRNA, the coding sequences for Cas9, BE3, and BE3-Y130F were first amplified from the plasmids using primers detailed in Appendix A. Subsequently, the resulting amplicons were used as templates for in vitro transcription using the mMESSAGE mMACHINE™ T7 ULTRA Transcription Kit (Thermo Fisher Scientific, Waltham, MA, USA, #AM1345) according to the manufacturer’s protocol. The synthesized mRNA was then purified and its concentration measured.

### 2.2. Production of Bovine Zygotes and Cytoplasmic Injection

Commercial bovine cumulus-oocyte complexes (COCs) were purchased from Simplot and shipped to ONPRC overnight via FedEx. For IVF, matured COCs were co-incubated with frozen-thawed spermatozoa in BO-IVF medium (IVF Bioscience, Falmouth, UK) for 9–12 h at 38.5 °C in a humidified atmosphere of 5% CO_2_. Frozen sperm was purchased from Select Sires Member Cooperative. Approximately 12 h post-insemination, presumptive zygotes were denuded of cumulus cells by gentle pipetting in TALP-Hepes medium and then washed. Zygotes were transferred to a microinjection dish containing droplets of TALP-Hepes medium under mineral oil. Zygotes were injected with Cas9 mRNA and sgRNA to assess the efficiency of the sgRNA. After confirmation of sgRNA efficiency, BE3 or BE3-Y130F was introduced into the cytoplasm of zygotes with sgRNA to examine the C-to-T conversion efficiency and window. Microinjection was performed using an inverted microscope (Nikon, Melville, NY, USA) equipped with micromanipulators (Narishige, Amityville, NY, USA). A mixture containing base editor mRNA (BE3 or BE3-Y130F at 100 ng/µL) and sgRNA (50 ng/µL) in RNase-free water was microinjected into the cytoplasm of each zygote. After microinjection, the embryos were washed and cultured in vitro in BO-IVC medium (IVF Bioscience) at 38.5 °C in a humidified atmosphere of 5% CO_2_ 5% O_2_, and 90% N_2_. Embryo development was monitored, and embryos were typically cultured for 6–7 days to reach the blastocyst stage.

### 2.3. Editing Efficiency Confirmation

Genomic DNA was extracted from individual blastocysts and arrested embryos (>8-cell stage). To prevent contamination from sperm DNA, the zona pellucida was removed by brief incubation in a pH 1.2 acidic solution. Whole genome amplification (WGA) was performed on each embryo using the REPLI-g Single Cell Kit (Qiagen, Santa Clarita, CA, USA, #150343). The WGA product was diluted 1:20 and used as a template for subsequent PCR analysis. *MYO7A* exon 4 was amplified using primers that correspond to regions in the flanking introns (Forward: 5′-GGAGTCAACCTGTTAGAACTGATTGTGTTTG-3′; Reverse: 5′-CATCCAGAACAAAAGAGAAAATAAGCCACAGACAG-3′) (Figure 1). PCR was performed using Phusion™ High-Fidelity DNA Polymerases (Thermo Fisher Scientific, F530S) under the following conditions: an initial denaturation at 98 °C for 3 min; followed by 31 cycles of 98 °C for 30 s, 64 °C for 30 s, and 72 °C for 30 s; and a final extension at 72 °C for 5 min. The PCR products were purified using a Thermo Scientific kit (K0701). The purified amplicons were then analyzed by Sanger sequencing to identify random indel mutations or C-to-T conversions resulting from Cas9 mRNA or base editor injection, respectively. Sequencing results were compared with the reference sequence using BLAST at NCBI (V2.17.0). The editing efficiency was confirmed by detecting random insertions or deletions introduced by the Cas9 mRNA. Additionally, C-to-T conversion efficiency and the editing window were evaluated in embryos injected with BE3 or BE3-Y130F.

## 3. Results

### 3.1. Editing Efficiency of sgRNA with Cas9 Microinjection

To validate the efficacy of the selected sgRNA, microinjection of Cas9 mRNA and the sgRNA was performed on 95 bovine zygotes (Table 1). After microinjection, a total of 6 blastocysts and 2 arrested embryos were obtained and subjected to WGA for PCR. Among the 8 genomic DNA samples, one arrested embryo did not yield a PCR band, indicating a failure to obtain genomic DNA from that embryo. The remaining PCR amplicons were purified and sequenced. Sanger sequencing results showed that all 6 blastocysts and one arrested embryo carried mutations in *MYO7A* exon 4, and the wild-type (non-mutated) allele was not detected (Figure 2). These results indicate that the sgRNA had high editing efficiency in bovine embryos and was suitable for use in BE3 and BE3-Y130F experiments.

### 3.2. C-to-T Conversion Efficiency and Window Comparison

Cytoplasmic microinjection of BE3 or BE3-Y130F mRNA, along with sgRNA, was successfully performed and repeated 3 times. For the BE3 test, a total of 143 zygotes were used, resulting in 11 blastocysts and 3 arrested embryos. For the BE3-Y130F experiment, a total of 171 zygotes were used, yielding 16 blastocysts and 3 arrested embryos for analysis (Table 2). Blastocysts and arrested embryos from both groups were subjected to WGA and PCR. All WGA samples successfully produced PCR products, and Sanger sequencing was conducted to confirm the C-to-T conversion efficiency and window.

Both BE3 and BE3-Y130F showed C-to-T conversion at cytosine positions 2, 3, 5, 7, and 8 in the sgRNA target sequence, measured from the end distal to the PAM (Figure 3 and Figure 4). This efficiency was defined as the percentage of analyzed embryos carrying at least one allele with the intended C-to-T conversion in any of the cytosines within the sgRNA target region. Comparing the C-to-T conversion efficiency between BE3 and BE3-Y130F, the sequencing results were classified into three categories: complete conversion (CC), incomplete conversion (IC), and no conversion (NC). In the case of CC, there was no cytosine detected, and only thymine was present. When both cytosine and thymine were detected, the result was classified as IC. If no cytosine-to-thymine conversion was observed, it was defined as NC. The BE3 group showed the highest CC rate of 85.71% at cytosine position 7. NC was only detected in one blastocyst (7.69%) at cytosine position 2. Notably, off-target C-to-T conversions, up to 7 bp outside the sgRNA sequence, were observed in 6 blastocysts (blastocysts # 1, 3, 5, 7, 8, and 9). Blastocyst # 3 and 7 showed CC outside of the sgRNA target region. In the BE3-Y130F group, the highest CC rate was 63.16% at cytosine positions 5 and 8. However, NC was observed at cytosine position 2 (38.89%) and position 3 (11.11%) (Figure 5). Similarly to the BE3 group, five blastocysts in the BE3-Y130F group showed off-target C-to-T conversions 2 bp downstream from the sgRNA sequence, but there was no complete conversion in the off-target sites identified. Detailed sequencing results are summarized in Appendix A.

All five target cytosines within the guide sequence were converted to thymine in both groups. In the BE3 group, blastocysts #3, 5, 7, and 8 each exhibited complete C-to-T conversion at all five positions. In the BE3-Y130F group, only blastocyst #7 showed complete conversion of all five cytosines to thymine (Appendix A). Additionally, random insertions or deletions (indels) were detected in one blastocyst in each group. C-to-T conversion was detected from the complementary DNA strand on off-target position 4 from the BE3 injection group in blastocyst #5.

## 4. Discussion

This study presents a direct analysis of the BE3 and BE3-Y130F cytosine base editors targeting the *MYO7A* gene in a bovine zygote, highlighting the distinctions in their C-to-T conversion characteristics, despite both editing tools achieving 100% overall conversion efficiency. These findings demonstrate that while both editors were highly effective, their respective editing windows and conversion consistency characteristics are distinct.

A key finding of this study is that BE3 exhibited a more consistent and predictable C-to-T conversion pattern compared to BE3-Y130F. In the BE3-injected group, conversions were most reliably observed at cytosine positions 5, 7, and 8 within the sgRNA target sequence, with a high CC rate of 85.71% at position 7. This defined and active editing window aligns with previous characterizations of BE3, which typically show peak activity at positions 4–8 relative to the protospacer sequence [6]. This consistency makes BE3 a robust tool for applications where the goal is to introduce a premature stop codon, which requires efficient conversion of any of several cytosines within a specific codon to generate TAA, TAG, or TGA sequences [12]. The broader activity window of BE3 increases the probability of successfully creating a stop codon and achieving a functional gene knockout. Moreover, a wider conversion window may give flexibility in targeting sites for premature stop codon production.

In contrast, the BE3-Y130F editor demonstrated a significantly narrower effective editing window. This was evidenced by the high frequency of NC events observed at positions 2 (38.89%) and 3 (11.11%), which BE3 readily edited. While BE3-Y130F showed high conversion efficiency at positions 5 and 8, its inactivity at the PAM-distal end of the protospacer suggests a more restricted operational range [13,14]. This narrower window, coupled with a reduced off-target conversion (2 bp vs. up to 7 bp outside the sgRNA for BE3), suggests that BE3-Y130F could be the superior choice for applications requiring high precision, such as gene therapy. For correcting a pathogenic single-nucleotide polymorphism (SNP), minimizing bystander edits at nearby cytosines is critical to avoid introducing unintended mutations [15]. The characteristics of BE3-Y130F, therefore, align well with the requirements for precise, single-base correction, even if it means sacrificing editing capability at specific positions within the target sequence. The comparative analysis of BE3—the most popular CBE for versatile zygote applications—and its refined Y130F variant, optimized for superior efficiency (e.g., 90% C-to-T rates), addresses a key gap in preclinical models. Such information will guide the development of strategies for large animal models of human diseases, such as for Usher Syndrome [16,17]. Although our findings provide useful insights into the performance of BE3 and BE3-Y130F at the *MYO7A* locus, a clinically relevant target for modeling Usher syndrome, it is difficult to draw broad conclusions about their overall efficiency and editing characteristics based on a single genomic site. It is acknowledged that assessing outcomes in a single gene target is a weakness of the current study. *MYO7A* is an important, clinically relevant target, and our findings support evaluating each gene of interest to understand differences in editing outcomes driven by their unique characteristics (e.g., chromatin structure, sequence-related features, and homology to other regions of the genome). As we evaluate additional target loci for the development of disease models or potential therapeutics, we will obtain more information that may yield a generalized summary of the editing outcomes of different base editors.

Our results also suggest potential differences in the temporal dynamics and cellular effects of the two editors. In the BE3 group, we observed that incomplete conversions were more frequently detected in arrested embryos compared to blastocysts. While the small sample size of arrested embryos precludes a definitive conclusion, this observation raises the hypothesis that BE3 may have editing activity continually during embryo development. This prolonged activity could lead to an accumulation of edits, potentially resulting in a CC on target sequences. The lower frequency of ICs in viable blastocysts suggests that in successful editing events, the C-to-T conversion process is completed efficiently and early. For these C-to-T conversion analyses, Sanger sequencing was performed without TA cloning, which still allowed a detailed assessment of allele-specific editing efficiency because the Sanger chromatograms clearly distinguished CC, IC, and NC. In future studies, TA cloning–based genotyping will be applied to more precisely evaluate mosaicism.

A low level of indel mutations was found in both groups (one blastocyst each), which is noteworthy. Even though base editors are designed to avoid double-strand breaks (DSBs) by using Cas9 nickase, a small number of indels can still occur, as previously reported. One common explanation is related to the base excision repair (BER) pathway in cells. When the deaminase changes the target cytosine (C) to uracil (U), the UGI part of the base editor is intended to prevent the uracil DNA glycosylase (UDG) from cutting out the uracil. But if the UGI does not function efficiently, UDG can remove the uracil, leaving an abasic (AP) site. An AP endonuclease can then cut this site. If this happens, together with the nick made by Cas9n on the other DNA strand, it can cause a temporary DSB. The cell then repairs this break via the error-prone non-homologous end joining (NHEJ) system, which can create indels [18,19]. However, this still occurs much less frequently than with normal CRISPR-Cas9, which is a significant benefit of base editors. Additionally, a rare C-to-T conversion, referred to as bystander deamination, was identified on the opposite DNA strand in the BE3 group. This unusual event is a rare but previously documented phenomenon in base editing.

While base editors minimize DSB-related toxicity, their impact on bovine embryo development warrants evaluation for transfer and pregnancy rates. Our blastocyst formation rates (7.7–9.4%) were similar to Cas9 controls (6.3%), and recent CBE studies have shown no significant reduction compared to non-injected/control groups [20,21]. Low yields likely reflect initial protocol optimization; future refinements (e.g., timing/injection material concentration) will enhance rates for nonhuman primate applications.

In this study, we did not perform off-target validation because outbred livestock species, such as pigs and bovines, exhibit high genomic diversity. More than 8 million single-nucleotide polymorphisms (SNPs) and 2.5 million small indels have been reported in these species, making it challenging to identify CRISPR-induced off-target events accurately [4,22].

This study demonstrates that both BE3 and BE3-Y130F are efficient base editors in the context of post-fertilization (i.e., zygote) cytoplasm microinjection. BE3, with its consistent and relatively broad editing window, is a reliable tool for gene disruption via premature stop codon introduction. BE3-Y130F, with its narrower editing window and reduced bystander activity, presents a more suitable option for precise gene correction paradigms, such as those envisioned for gene therapy. This direct comparison provides valuable guidance for researchers in selecting the optimal base editing tool to achieve specific genetic outcomes in large animal models and beyond.

## Figures and Tables

**Figure 1 cimb-47-01033-f001:**
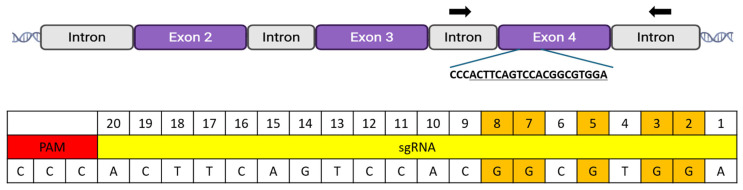
*MYO7A* target exon and sgRNA sequence. To evaluate BE3 and BE3-Y130F, an sgRNA was designed to target exon 4 of *MYO7A*. The target sequence was selected to assess C-to-T conversion efficiencies and positional editing preferences, recognizing that it would not generate a premature stop codon. The guide sequence was chosen because it contained five guanines at positions 2, 3, 5, 7, and 8. Since the guide sequence was in the reverse orientation, these guanines would be converted to thymine by BE3 or BE3-Y130F. Black arrows indicated the primer binding sites in the flanking region of the adjacent introns that allow for the amplification of the entire exon 4 sequence.

**Figure 2 cimb-47-01033-f002:**
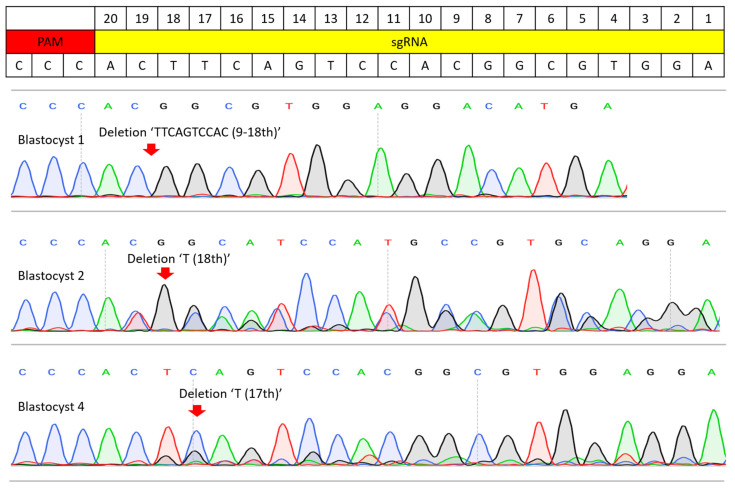
Sanger sequencing results of the sgRNA efficiency test. PCR amplicons from the injection of Cas9 mRNA and sgRNA were analyzed by Sanger sequencing. Blastocyst 1 resulted in a homozygous mutation and a 10 bp deletion (9–18th base pair of sgRNA). Blastocysts 2 and 4 possessed mosaic mutations, but no wild-type sequence was present. Blastocyst had 18th ‘T’ deletion and blastocyst 4 had 17th ‘T’ deletion. Red arrows indicate where the edited sequence diverges from the wild-type sequence.

**Figure 3 cimb-47-01033-f003:**
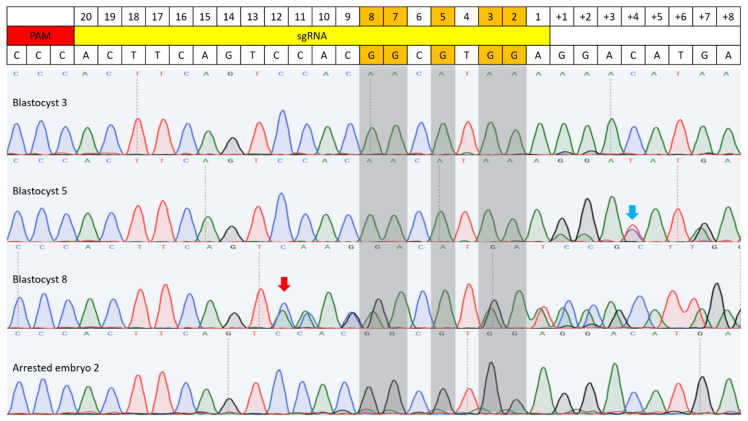
C-to-T base editing in BE3-injected bovine embryos. Sequencing results of four embryos are included. Blastocyst 3 showed complete C-to-T conversion at both the target site and an off-target site, with the off-target conversion occurring 7 bp downstream of the sgRNA sequence. Blastocyst 5 also exhibited complete conversion at the target site, and C-to-T conversion was observed on the complementary DNA strand at off-target position 4 (blue arrow). Blastocyst 8 displayed a random indel mutation, with the mutation start point indicated by a red arrow. Arrested embryo 2 showed an example of incomplete conversion. Gray-highlighted bases mark the expected G-to-A conversion sites.

**Figure 4 cimb-47-01033-f004:**
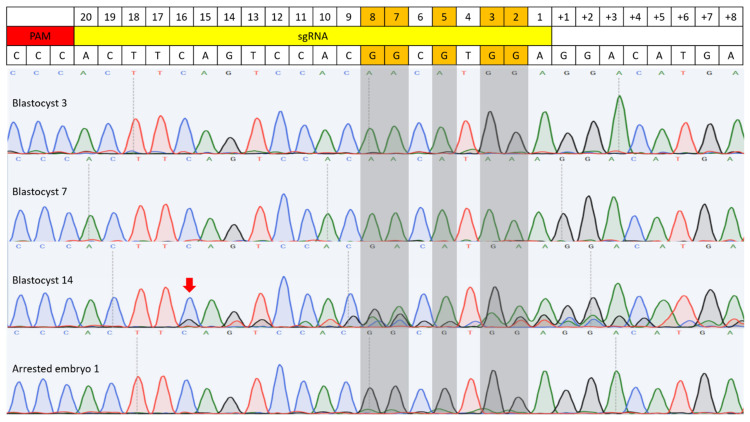
C-to-T base editing in BE3-Y130F injected embryos. Sequencing results of four embryos are included. Blastocysts 3 and 7 were examples of C-to-T conversion with the sgRNA target sequence. A random indel mutation was detected in blastocyst 14 (Red arrow). Arrested embryo 2 showed an example of incomplete conversion. Gray-highlighted bases mark the expected G-to-A conversion sites.

**Figure 5 cimb-47-01033-f005:**
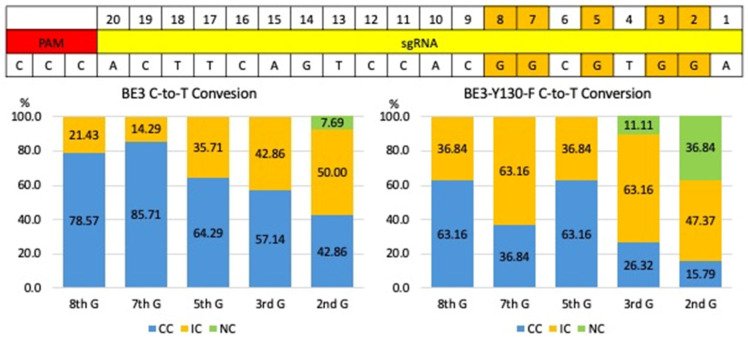
Comparison of C-to-T base editing efficiency and editing window of BE3 and BE3-Y130F. The graphs showed the frequencies of C-to-T conversion outcomes for BE3 (**left**) and the BE3-Y130F (**right**) on five specified target guanines at positions 2, 3, 5, 7, and 8. The graph represents the percentage of conversions for three different categories: completely converted (CC), incomplete conversion (IC), and no conversion (NC). CC: targeted G was completely converted A (blue); IC: both targeted G and converted A detected (Orange); NC: wild-type sequence (green).

**Table 1 cimb-47-01033-t001:** Embryo development outcomes: assessing *MYO7A* exon 4 sgRNA efficiency.

Injection Material	# of Injected Zygotes	# of Blastocysts	# of Arrested Embryos Analyzed	Results	
				# of Knockout Embryos	# of Embryos: Wild Type
Cas9 mRNA + sgRNA	95	6	2	7	0

**Table 2 cimb-47-01033-t002:** Embryo development outcomes following BE3 and BE3-Y130F injection at the zygote stage.

Injection Material	# of Injected Zygotes	# of Blastocysts	# of Arrested Embryos Analyzed	Results	
				# with a C-to-T Conversion	# with no Conversion
BE3 mRNA + sgRNA	143	11	3	14	0
BE3-Y130F + sgRNA	171	16	3	19	0

## Data Availability

The original contributions presented in this study are included in the article/Appendix A. Further inquiries can be directed to the corresponding author.

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
