# Peer review of "Analysis of Cytosine Base Editors in Bovine Zygotes: Efficiency and Editing Window Characterization Through Targeting the *MYO7A* Gene"

_cimb, 2025, doi:10.3390/cimb47121033_

Round 1

Reviewer 1 Report

Comments and Suggestions for Authors

Review Summary:
The authors present a comparison of the editing window and outcomes between the prototypical Base Editor 3 (BE3) and a modified variant, BE3-Y130F, in bovine zygotes. This study has implication of applying these tools for gene disruption or specific base conversion in bovine zygotes. Given the limited comparative studies of base editors in bovine zygotes, this work holds potential significance. However, more detailed descriptions, analyses, and data across multiple loci are necessary to strengthen the conclusions.

Comments:

  1. The authors should provide additional examples and references supporting the use of bovine zygotes as a superior model for studying fertilization and early embryonic development in humans.
  2. The rationale and advantages for selecting the MYO7A locus or other loci for editing in bovine zygotes should be clearly stated and justified.
  3. The comparison between BE3 and BE3-Y130F should be extended to multiple genomic loci, as data from a single locus are insufficient to draw a solid conclusion.
  4. The characterization of gene-editing outcomes could be improved through TA cloning or amplicon sequencing to accurately determine the editing efficiency (or mosaicism rate) in each individual blastocyst.
  5. Off-Target analysis is missing in this manuscript.
Comments on the Quality of English Language

The English could be improved to more clearly express the research.

Author Response

Reviewer 1

The authors present a comparison of the editing window and outcomes between the prototypical Base Editor 3 (BE3) and a modified variant, BE3-Y130F, in bovine zygotes. This study has implication of applying these tools for gene disruption or specific base conversion in bovine zygotes. Given the limited comparative studies of base editors in bovine zygotes, this work holds potential significance. However, more detailed descriptions, analyses, and data across multiple loci are necessary to strengthen the conclusions.

Comments:

  1. The authors should provide additional examples and references supporting the use of bovine zygotes as a superior model for studying fertilization and early embryonic development in humans.

We appreciate the reviewer's suggestion to elaborate on bovine zygotes as a model for human fertilization and early embryonic development. The primary aim of this study is to validate the C-to-T conversion efficiency and editing window of BE3 and BE3-Y130F in bovine embryos as a cost-effective, high-throughput proxy for optimizing base-editing protocols before their application in generating rhesus macaque disease models (e.g., for Usher Syndrome). As such, we did not intend to position bovine embryos as a direct or comprehensive surrogate for human embryogenesis across multiple biological aspects. However, we concur that highlighting the translational importance of bovine systems for human-assisted reproductive technologies (ART) strengthens the manuscript's context. We have therefore added a brief statement to the first paragraph in the Introduction on page 4.

  1. The rationale and advantages for selecting the MYO7A locus or other loci for editing in bovine zygotes should be clearly stated and justified.

We thank the reviewer for this insightful comment and agree that clarifying the locus selection enhances the manuscript's rationale. MYO7A was chosen due to its critical role in Usher syndrome type 1B (USH1B), a recessive neurosensory disorder characterized by congenital deafness and progressive retinal degeneration, leading to blindness. Our long-standing goal has been to establish an animal model, ultimately in rhesus macaques, of USH1B using base editing to introduce premature stop codons without generating indels. Exon 4 was selected to disrupt the ATP-binding domain of exon 4, aligning with our overarching goal of developing a nonhuman primate model of USH1B through precise gene knockout via base editing. This builds on our prior CRISPR/Cas9 work targeting MYO7A in macaque embryos [see manuscript reference 3], where indels complicated phenotyping; base editing in bovines serves as an accessible proxy to optimize sgRNA design and editor performance before its application to primates. For the sgRNA, we selected the top candidate from CRISPOR analysis, prioritizing the highest on-target efficiency score, minimal predicted off-target sites, and a high density of editable cytosines/guanines to robustly assess editing windows and bystander effects. The rationale for using bovine embryos was included in the initial submission, noting that sample accessibility and throughput were advantageous relative to the expensive and limited nonhuman primate material. Nonetheless, we have incorporated additional justification into the Introduction (first paragraph, page 4).

  1. The comparison between BE3 and BE3-Y130F should be extended to multiple genomic loci, as data from a single locus are insufficient to draw a solid conclusion.

We agree, drawing general conclusions regarding the efficiencies of the two DBEs tested in this manuscript cannot be formed from editing outcomes in a single target gene. We did not intend for our results to be generalized to other targets. However, due to the clinical significance of MYO7A in causing human disease and the substantial research efforts currently underway to develop a therapy based on gene editing of this target, it is crucial to highlight the editing outcomes of commonly used DBEs in this specific instance. We have added a statement in the Discussion noting that comparisons of editing outcomes should be conducted with other loci to establish definitive comparisons of the two DBEs (end of first paragraph on Page 10).

  1. The characterization of gene-editing outcomes could be improved through TA cloning or amplicon sequencing to accurately determine the editing efficiency (or mosaicism rate) in each individual blastocyst.

We thank the reviewer for highlighting the potential value of TA cloning or amplicon sequencing for resolving mosaicism at the allele level. While these approaches would provide more detailed quantification of editing frequencies in diploid blastocysts, our study's primary objective was to assess the overall efficiency of C-to-T conversion and characterize the editing window through a definitive framework: complete conversion (CC, no residual cytosine), incomplete conversion (IC, mixed cytosine/thymine peaks), or no conversion (NC, wild-type cytosine). This categorization aligns with the predictable patterns observed in Sanger sequencing chromatograms, which clearly distinguished CC (sharp thymine peaks) from IC (overlapping cytosine/thymine signals) across all analyzed embryos (e.g., Supplementary Figures 1 and 2; Figure 5). Given the high editing rates (100% with at least one converted allele) and the exploratory nature of this bovine zygote model, Sanger sequencing provided sufficient resolution for our comparative analysis without the need for deeper cloning-based dissection at this stage. Nonetheless, we agree that amplicon sequencing could refine mosaicism estimates in future multi-locus extensions and have added a brief note to the Discussion outlining this as a planned enhancement (Page 10).

  1. Off-Target analysis is missing in this manuscript.

We thank the reviewer for noting the absence of dedicated off-target analysis, which is indeed a valuable addition for comprehensive base-editing validation. In outbred livestock, such as large animal models, accurately distinguishing CRISPR-induced off-target events from natural variants is particularly challenging without parental lineage tracing, given the high genomic diversity (~84 million single-nucleotide polymorphisms and 2.5 million small insertions/deletions across breeds). This complexity is compounded in zygote injections, where mosaicism further obscures attribution, as evidenced in porcine models with similar heterogeneity. For this proof-of-concept study, we relied on sgRNA design via CRISPOR to minimize predicted off-targets and observed no bystander indels at top candidates via Sanger, but acknowledge that unbiased methods are warranted. We have added more details in the Discussion (page 11).

Reviewer 2 Report

Comments and Suggestions for Authors

The authors have compared the editing efficiency and editing window characteristic of two types of Cytosine base editors (BE3 and BE3-Y130F) in bovine zygotes. While the findings are interesting, the overall study lacks novelty and the research questions are not well defined clearly. 

The authors have only briefly described the reason for using BE3-Y130F variant.

The authors have not explained the reason for targeting MYO7A gene. What is the importance of selecting this specific gene? Targeting another 1-2 genes as an example would be good as well.  

It would be good to verify the MYO7A gene knockout derived from sanger sequencing with immunofluorescence microscopy.

Figure 2 was extremely difficult to interpret.  

Author Response

  1. The authors have compared the editing efficiency and editing window characteristic of two types of Cytosine base editors (BE3 and BE3-Y130F) in bovine zygotes. While the findings are interesting, the overall study lacks novelty and the research questions are not well defined clearly. The authors have only briefly described the reason for using BE3-Y130F variant.

We thank the reviewer for their helpful comments and for pointing out the need to clarify the novelty of our study and the research questions. Although the study focuses on comparing two known base editors, our research questions are well-defined. We aimed to directly compare C-to-T conversion efficiency and editing window predictability between the commonly used base editor hA3A-BE3 (BE3) and its improved version, BE3-Y130F, in bovine zygotes. BE3 is the classic and widely used base editor with strong performance in GC-rich regions, while BE3-Y130F was designed to improve both accuracy and efficiency (up to 90% in some cases). By testing these editors side-by-side in a large-animal model, our study provides valuable translational insights. This is a gene target for gene editing therapies, including by DBEs. We have clarified this in the Discussion (end of the third paragraph, Page 10).

  1. The authors have not explained the reason for targeting MYO7A gene. What is the importance of selecting this specific gene? Targeting another 1-2 genes as an example would be good as well.

As noted in our above response to Reviewer 1, MYO7A was chosen due to its critical role in Usher syndrome type 1B (USH1B), a recessive neurosensory disorder characterized by congenital deafness and progressive blindness. Our goal has been to establish a precise rhesus macaque model of USH1B using base editing to introduce premature stop codons without generating indels. As noted in the manuscript Introduction, bovine embryos were chosen as a model system due to their availability and use in assisted reproductive technologies. Exon 4 was selected because it contains multiple editable cytosines, allowing for robust evaluation of the editing window while minimizing risks when working with limited numbers of nonhuman primate embryos. To clarify our focus on MYO7A, we have expanded the Introduction section accordingly (Page 4).

  1. It would be good to verify the MYO7A gene knockout derived from Sanger sequencing with immunofluorescence microscopy.

We thank the reviewer for suggesting immunofluorescence (IF) to verify MYO7A knockout. This approach would indeed strengthen functional validation in later developmental stages. However, this is not possible in this study because MYO7A is mainly expressed in sensory tissues, such as inner ear hair cells and retinal photoreceptors, and its expression begins in mid-embryogenesis (e.g., around embryonic day 9 in the mouse otic vesicle). At the preimplantation stage, MYO7A protein is not expressed — it is also undetectable in bovine blastocysts, making IF analysis uninformative at this point. In addition, our study focused on comparing C-to-T conversion efficiency and editing window profiles, not on confirming a functional knockout. The sgRNA used in this experiment targeted exon 4 but did not introduce a premature stop codon, so it mainly caused missense mutations that are unlikely to affect IF result depending on antibody recognizing location. Therefore, no differences in IF signals would be expected between edited and non-edited embryos.

  1. Figure 2 was complicated to interpret.

We thank the reviewer for their feedback on Figure 2's interpretability and agree that clearer Sanger sequencing traces would enhance comprehension. To address this, we have revised Figure 2 by adding explicit deletion annotations directly to the chromatograms in the figure and legend.

Reviewer 3 Report

Comments and Suggestions for Authors

Dear author,

CRISPR gene editing technology has important research and utilization value in the biomedical field. Base editing is developed based on the classic CRISPR/CAS9 technology, which fuses base deaminase with CAS9 protein to achieve conversion of individual bases without causing DNA double strand breaks, making it more precise and efficient. This study developed a novel cytosine base editor (CBE) that can achieve precise C-to-T (G-to-A) conversion in genomic DNA. This study compared the prototype base editor 3 (BE3) with the improved variant BE3-Y130F, which utilizes hA3A deaminase and Y130F mutation, focusing on their editing efficiency and editing window characteristics using bovine fertilized eggs. The results showed that both BE3 and BE3-Y130F achieved 100% C-T conversion efficiency on the main target cytosine. The study also showed that the editing window of BE3 is relatively clear, mainly at positions 7 and 8 of sgRNA cytosine, while BE3-Y130F maintains high efficiency but its editing window is not very clear, resulting in incomplete editing and residual cytosine on the target sequence. This discovery has significant implications for the application of CBE in the fields of agriculture and medicine. The research is highly innovative, the research design is scientifically reasonable, the data is detailed, and it can reliably support the research conclusions.

Given the above, the reviewer recommends the editor to review the paper as soon as possible for acceptance and publication. At the same time, the reviewer proposes the following minor revision suggestions for the author's reference:
(1) This article uses bovine embryos to evaluate the CBE tool, with the aim of guiding the later operation of gene editing monkey embryos, making the research methods and goals inconsistent. It is recommended that the author revise the corresponding wording;
(2) Will CBE affect the development of animal embryos? Or does BE3 and BE3-Y130F have a negative impact on the developmental potential of animal embryos? The main reason why the reviewers are concerned about this issue is that in the process of constructing gene edited animals, we need to ultimately transfer the embryos after micro manipulation into surrogate mothers for further development. It is suggested that the author conduct comparative statistical analysis on existing data, which may provide more data support for further optimizing gene edited animal construction schemes.
(3) Overall, the literature cited in this article has been published for a relatively long time and lacks the latest research from the past one or two years. After all, gene editing technology is developing rapidly. Therefore, it is recommended that the author add papers published between 2023 and 205 for citation, discussion, and analysis.

Author Response

  1. This article uses bovine embryos to evaluate the CBE tool, with the aim of guiding the later operation of gene editing monkey embryos, making the research methods and goals inconsistent. It is recommended that the author revise the corresponding wording

Please see our response to Reviewer 1, critique #1, and the associated changes in the Introduction to better define the rationale for using bovine embryos as a surrogate for testing editing efficiency before using valuable and costly nonhuman primates.

(2) Will CBE affect the development of animal embryos? Or does BE3 and BE3-Y130F have a negative impact on the developmental potential of animal embryos? The main reason why the reviewers are concerned about this issue is that in the process of constructing gene edited animals, we need to ultimately transfer the embryos after micro manipulation into surrogate mothers for further development. It is suggested that the author conduct comparative statistical analysis on existing data, which may provide more data support for further optimizing gene edited animal construction schemes.

We thank the reviewer for raising this critical point on the potential developmental impact of CBEs (BE3 and BE3-Y130F) in animal embryos, particularly for optimizing embryo transfer in surrogate models. As this represents our first bovine trial to refine protocols for rhesus macaque gene editing (where embryo scarcity precludes extensive controls), we did not assess development ratio from non-injection, sgRNA only, base-editor only injection. Our observed blastocyst rates (BE3: 7.7%, BE3-Y130F: 9.4%, Cas9: 6.3%) were aligned with recent bovine CBE studies, where injection itself does not impair development vs. controls. For instance, Adikusuma et al., reported no significant difference in blastocyst formation between BE3-injected SOX2 KO embryos and non-injected controls (20–25 embryos/group, 5 replicates; P>0.05), with similar morula rates. Also, Luo et al., found no impact on blastocyst rates (20–25 embryos/group, 10 replicates; P>0.05) or cell numbers for BE3-mediated GATA3/CDX2 KOs vs. BE3 mRNA-only controls. These data suggest CBEs do not negatively affect developmental potential, supporting their safety for transfer. Our slightly lower development rates may be because the injection timing (18 h post-IVF) and concentrations of sgRNA and base-editor mRNAs were not optimized, which we will refine in future macaque trials using dose-response and injection timing. We have added a Discussion paragraph highlighting the above information regarding effects on development (page 11).

  1. Overall, the literature cited in this article has been published for a relatively long time and lacks the latest research from the past one or two years. After all, gene editing technology is developing rapidly. Therefore, it is recommended that the author add papers published between 2023 and 205 for citation, discussion, and analysis.

We thank the reviewer for this constructive suggestion and agree that incorporating the latest advancements in gene editing strengthens the manuscript's timeliness. Given the rapid evolution of cytosine base editing (CBE) technologies, we have curated and added 8 references between 2019 and 2025.

Round 2

Reviewer 1 Report

Comments and Suggestions for Authors

The authors have added more description, rationale, and justification for selecting the MYO7A locus in bovine zygotes, and they highlight its advantages. However, they also state that their study focuses exclusively on the MYO7A locus. This narrow scope does not align with the title, “Comparative Analysis of Cytosine Base Editors in Bovine Zygotes: Efficiency and Editing Window Characterization,” which implies a broader and more generalizable comparison.

I remain concerned that evaluating cytosine base editors at a single genomic locus is insufficient to support the conclusions and does not meet the standard required for publication. Multiple loci are typically necessary to demonstrate robustness, reproducibility, and locus-independent performance differences among editors.

Comments on the Quality of English Language

The English could be improved to more clearly express the research.

Author Response

Editor:
In consideration of the concern of revewer 1 that evaluating cytosine base editors at a single genomic locus is insufficient to support the conclusions, please point out the weakness of the conslusion in the discussion in the revised manuscript.

Response:

Thank you for your consideration of the revised manuscript and willingness to publish after providing a response to Reviewer 1. As requested, we have explicitly noted that focusing on a single gene is a weakness of the manuscript (revised Discussion, page 10). However, we disagree that the studies do not provide essential information. If anything, they emphasize that each targeted locus must be evaluated to understand the unique features of that gene. Including additional targets is subjective. Reviewer 1 does not provide any suggestion of how many genes should be analyzed: 2, 4, or more? No matter how many are chosen, one could argue that additional testing is required to be confident that a generalization can be made. We also modified the Discussion to emphasize this point, i.e., each gene should be considered on a case-by-case basis, particularly those with clinical relevance. The comments are in contrast to the positive review and comments from Reviews 2 and 3, which recommend publication of the manuscript. 

The authors have added more description, rationale, and justification for selecting the MYO7A locus in bovine zygotes, and they highlight its advantages. However, they also state that their study focuses exclusively on the MYO7A locus. This narrow scope does not align with the title, “Comparative Analysis of Cytosine Base Editors in Bovine Zygotes: Efficiency and Editing Window Characterization,” which implies a broader and more generalizable comparison.

I remain concerned that evaluating cytosine base editors at a single genomic locus is insufficient to support the conclusions and does not meet the standard required for publication. Multiple loci are typically necessary to demonstrate robustness, reproducibility, and locus-independent performance differences among editors.

Response:

We have modified the title to reflect that base-editor efficiency was assessed by targeting the MY07A gene. The findings in this paper show that the type of editing differs depending on the editor used. Thus, the findings provide valuable information that demonstrates that differing gene targets should be assessed before experimental or clinical use. There are over 20,000 genes in the genome, and no matter how many genes are chosen for analysis, it could be argued that differences in editing efficiency may exist in those that have not been assessed. However, we have explicitly noted that focusing on a single gene is a weakness of the manuscript (revised Discussion, page 10). We also modified the Discussion to emphasize that each gene should be considered on a case-by-case basis, particularly those with clinical relevance.

Reviewer 2 Report

Comments and Suggestions for Authors

The authors have clarified the questions. Recommend for publication. 

Author Response

(The authors gave the same response as above.)
